# Favipiravir-resistant influenza A virus shows potential for transmission

**Daniel H. Goldhill**[1], **Ada Yan**[2], **Rebecca Frise**[1], **Jie Zhou**[1], **Jennifer Shelley**[1], **Ana Gallego Cortés**[1], **Shahjahan Miah**[3], **Omolola Akinbami**[3], **Monica Galiano**[3¤], **Maria Zambon**[3], **Angie Lackenby**[3], **Wendy S. Barclay**[1]*

**1** Department of Infectious Disease, Imperial College, London, United Kingdom, **2** Department of Infectious Disease Epidemiology, Imperial College, London, United Kingdom, **3** Public Health England, London, United Kingdom

¤ Current address: Worldwide Influenza Centre, The Francis Crick Institute, London, United Kingdom
* w.barclay@imperial.ac.uk

**Data Availability Statement:** Sequencing data can be found at https://www.ebi.ac.uk/ena (project number PRJEB39934.) All other relevant data are

## Abstract

Favipiravir is a nucleoside analogue which has been licensed to treat influenza in the event of a new pandemic. We previously described a favipiravir resistant influenza A virus generated by in vitro passage in presence of drug with two mutations: K229R in PB1, which conferred resistance at a cost to polymerase activity, and P653L in PA, which compensated for the cost of polymerase activity. However, the clinical relevance of these mutations is unclear as the mutations have not been found in natural isolates and it is unknown whether viruses harbouring these mutations would replicate or transmit in vivo. Here, we infected ferrets with a mix of wild type p(H1N1) 2009 and corresponding favipiravir-resistant virus and tested for replication and transmission in the absence of drug. Favipiravir-resistant virus successfully infected ferrets and was transmitted by both contact transmission and respiratory droplet routes. However, sequencing revealed the mutation that conferred resistance, K229R, decreased in frequency over time within ferrets. Modelling revealed that due to a fitness advantage for the PA P653L mutant, reassortment with the wild-type virus to gain wild-type PB1 segment in vivo resulted in the loss of the PB1 resistance mutation K229R. We demonstrated that this fitness advantage of PA P653L in the background of our starting virus A/England/195/2009 was due to a maladapted PA in first wave isolates from the 2009 pandemic. We show there is no fitness advantage of P653L in more recent pH1N1 influenza A viruses. Therefore, whilst favipiravir-resistant virus can transmit in vivo, the likelihood that the resistance mutation is retained in the absence of drug pressure may vary depending on the genetic background of the starting viral strain.

## Author summary

In the event of a new influenza pandemic, drugs will be our first line of defence against the virus. However, drug resistance has proven to be particularly problematic to drugs against influenza. Favipiravir is a novel drug which might be used against influenza virus in the event of a new pandemic. Is resistance likely to be a problem for the use of

within the manuscript and its Supporting Information files.

**Funding:** Funding was provided by Wellcome Trust Grant 205100 (DG, WB) and Wellcome Trust Grant 200187/Z/15/Z (AY, RF, JZ, WB). (wellcome.ac.uk) The funders had no role in study design, data collection and analysis, decision to publish, or preparation of the manuscript.

**Competing interests:** I have read the journal's policy and the authors of this manuscript have the following competing interests: WB has received honoraria from Roche, Sanofi Pasteur and Seqirus. The rest of the Authors have nothing to declare.

favipiravir? Our previous work has shown that resistance to favipiravir can be generated in cell culture but we don't know whether there will be a cost preventing the spread of resistance in whole organisms. Here, we used a mix of wild-type and resistant influenza viruses from early in the 2009 pandemic to test whether viruses resistant to favipiravir could transmit between ferrets. We found that the resistant viruses could transmit but that the resistance mutation was selected against within some ferrets. Using modelling and in vitro experiments, we found that the resistant mutation was selected against in the influenza strain from our experiment but not in more recently evolved strains. Our results show that favipiravir resistant viruses could spread if resistance is generated but the probability will depend on the genetic background of the virus.

## Introduction

Influenza virus is a negative strand virus that causes significant morbidity and mortality worldwide. Like most other RNA viruses, influenza virus has a fast rate of evolution which allows it to evolve in response to immune pressure as well as antiviral drugs [1–3]. Antiviral resistance has been a major problem limiting the effectiveness of antiviral drugs against influenza [4,5]. Significant resistance has evolved against the two main classes of antiviral drugs, adamantanes and neuraminidase inhibitors, which have been used clinically against influenza [5–8]. Resistance can also evolve to baloxavir, a recently approved drug that inhibits the cap-snatching ability of the polymerase [9]. When developing new drugs, it is vital to understand both the ease with which resistance evolves and the likelihood that resistant variants will transmit especially in the absence of drug pressure. This will inform whether a new drug can be effectively used against influenza on a global scale.

Favipiravir is a novel antiviral drug licensed in Japan for the treatment of influenza in the event of a new pandemic [10–12]. Favipiravir is a nucleoside analogue which targets the influenza polymerase and acts as a mutagen [13–15]. Favipiravir is active against influenza A and B viruses including strains that are resistant to other classes of antiviral drugs [10,16]. Previously, we demonstrated that Influenza A/England/195/2009 (Eng195), an early isolate from the 2009 H1N1 pandemic, could evolve resistance to favipiravir [17]. We showed that two mutations, K229R in PB1 and P653L in PA, were needed to evolve resistance [17]. K229R provided resistance to favipiravir at a cost to polymerase activity which was compensated by P653L. K229R has not been found in any natural pH1N1(2009) isolates, which is unsurprising as favipiravir has not been widely used to treat influenza cases. The P653L mutation has also not been found in any natural pH1N1(2009) isolates, which may suggest that it does not confer a fitness advantage in vivo. Although the P653L fully compensated for the cost to polymerase activity and virus replication in vitro, it is unknown whether the resistant virus would replicate in vivo or transmit whilst maintaining resistance.

Transmission studies in animal models such as the ferret have revealed whether drug resistant influenza viruses have fitness costs and can help inform about the likelihood of the emergence of resistance [18–20]. Early studies with the adamantane, rimantadine, suggested that there was no fitness cost preventing transmission of resistant influenza in a household setting [21] and resistance has indeed become widespread. Oseltamivir resistant H1N1 viruses were originally shown to be unlikely to transmit [22] but around 2007 additional mutations in the N1 neuraminidase emerged that were subsequently shown to affect NA such that the resistance mutation actually conferred a fitness advantage [5] and widespread resistance to oseltamivir ensued [23]. More recently, oseltamivir resistant pH1N1(2009) [24] and baloxavir resistant

H3N2 influenza A virus have been shown to transmit between ferrets without a fitness cost [25,26]. In humans so far, resistance to these drugs appears mostly limited to treated patients or small outbreaks [27,28] but these animal transmission studies suggest that continued use of monotherapy may lead to more widespread resistance.

Favipiravir resistance is only likely to develop in an individual treated with favipiravir [17]. However, if a favipiravir resistance mutation could transmit and replicate without a fitness cost, resistance could gradually spread throughout an untreated population. This could be especially problematic if initial transmission happened in a hospital environment or if resistance emerged on a background where influenza was resistant to other antivirals. Although, favipiravir treatment is unlikely to be common enough to select for resistance at the population level, previous drug resistance mutations have spread in the absence of extensive drug use [23]. It is possible that through drift, or being associated with other beneficial mutations, favipiravir resistance could spread through the population in the absence of a cost to resistance. Therefore, it is important to determine whether there is a cost to favipiravir resistance in vivo. Interestingly, K229 is a conserved residue in the F-motif of the polymerase [17] and favipiravir resistant chikungunya virus containing the corresponding mutation to K229R in influenza, has been shown to reproduce less efficiently in mosquitos which led to slower transmission [29]. This fitness cost in mosquitos was unexpected as there was no difference in fitness between favipiravir resistant virus and wild-type virus in mammalian cell culture [29,30]. In the present study, we infected ferrets with a mixture of wild-type and resistant virus and tested whether resistant virus would transmit or be outcompeted by the wild-type virus. We constructed a simple model to explain the changes in genotype frequencies observed in our experiment. Finally, we compared pH1N1 PA sequences from 2009–2010 to test whether our results were contingent on the genetic background of the virus.

## Results

### Favipiravir resistant virus transmits between ferrets

To test whether favipiravir resistant virus could transmit by direct or indirect contact, we inoculated ferrets with a mix of resistant virus bearing PB1 K229R + PA P653L and the corresponding wild-type virus, Eng195, a prototypical first wave pH1N1 2009 virus. By inoculating with a mix of virus, we could investigate whether there was a detectable fitness difference between the favipiravir resistant and the wild-type virus. We used a low percentage of wild-type virus to maximize the probability of resistant virus transmitting in the event that there was a fitness cost to resistance in the ferrets. Four donor ferrets were inoculated with K229R + P653L and Eng195 viruses in the ratio of 95:5. After 24 hours, each donor ferret was housed with a direct contact sentinel ferret to measure contact transmission. In addition, an indirect contact sentinel animal was housed in a separate adjacent cage to measure airborne transmission.

All 4 donor ferrets were successfully infected and shed virus in the nasal wash with a peak viral titre on day 2 and a secondary peak for most donors on day 4 or 5 as has been seen previously for ferrets infected with this dose of pH1N1 virus [31] (Fig 1). All 4 direct contact sentinels became infected with the first positive nasal washes occurring between days 2–5. 3 of the 4 indirect contact sentinels became infected and their first positive nasal washes occurred between 3–7 days following infection of the donors. The direct contact and indirect contact sentinels had peak viral titres comparable to the donors suggesting that they were robustly infected.

Two time points were selected from the daily nasal washes collected from each ferret to sequence virus shed in the nasal washes by both whole genome sequencing and more targeted

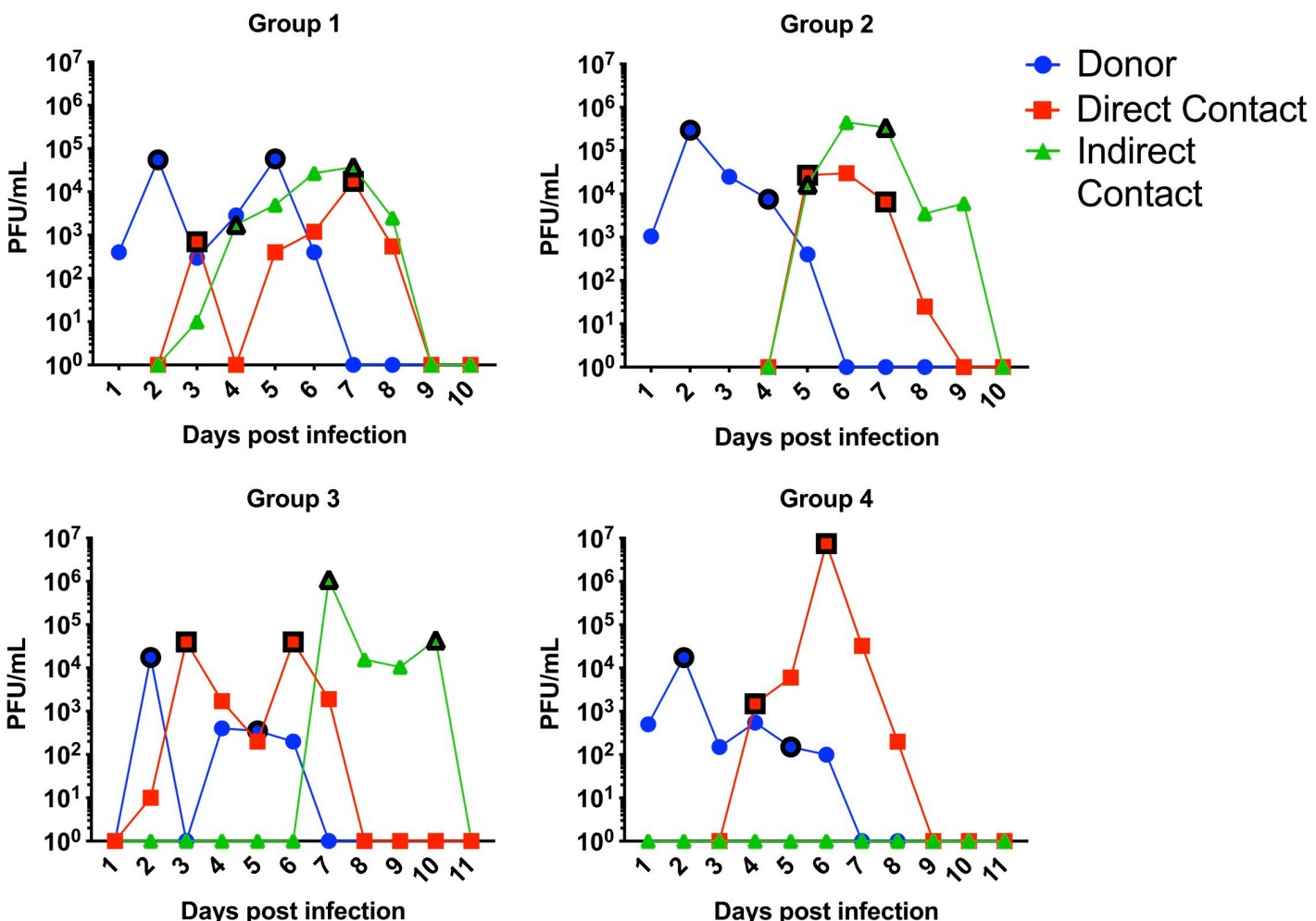

**Fig 1. 4 donor ferrets were infected with 10^4 PFU of a virus mix of wildtype Eng195 and K229R+P653L.** Direct contact and indirect sentinels were exposed from day 1. Ferrets were nasal washed each day and virus infectivity in nasal wash titred by plaque assay. 2 samples were chosen for sequencing from each ferret and are denoted by the black outlined symbols.

sequencing of the PB1 and PA segments (Fig 1). The first time point was either the first or second positive nasal wash to give a snapshot of the diversity of viruses shortly after infection. The second time point was chosen 2–4 days after to show how viral genotypes change within a host ferret over time. PB1 and PA sequencing revealed that RNA ratio in the inoculum was 95% K229R + P653L and 5% Eng195. Sequencing of virus in the nasal wash showed high levels of both R229 in PB1 (>50%) and L653 in PA (>85%) in all infected ferrets indicating that the K229R + P653L virus could productively infect ferrets and was transmitted both through direct and indirect contact transmission routes (Fig 2). The earliest sample after acquisition of virus in 2 of 4 contact ferrets and 2 of 3 indirect contact ferrets contained 100% K229R + P653L, with no transmission of any wild-type segments. This demonstrated that favipiravir resistant virus could transmit between ferrets in the absence of drug pressure.

## Whole genome sequencing reveals no additional changes in PB1 and PA

Viruses shed in nasal wash underwent whole genome sequencing to search for additional mutations which might be required for efficient transmission or to further compensate for K229R or P653L. We found no additional mutations in PB1 or PA in any of the donor ferrets

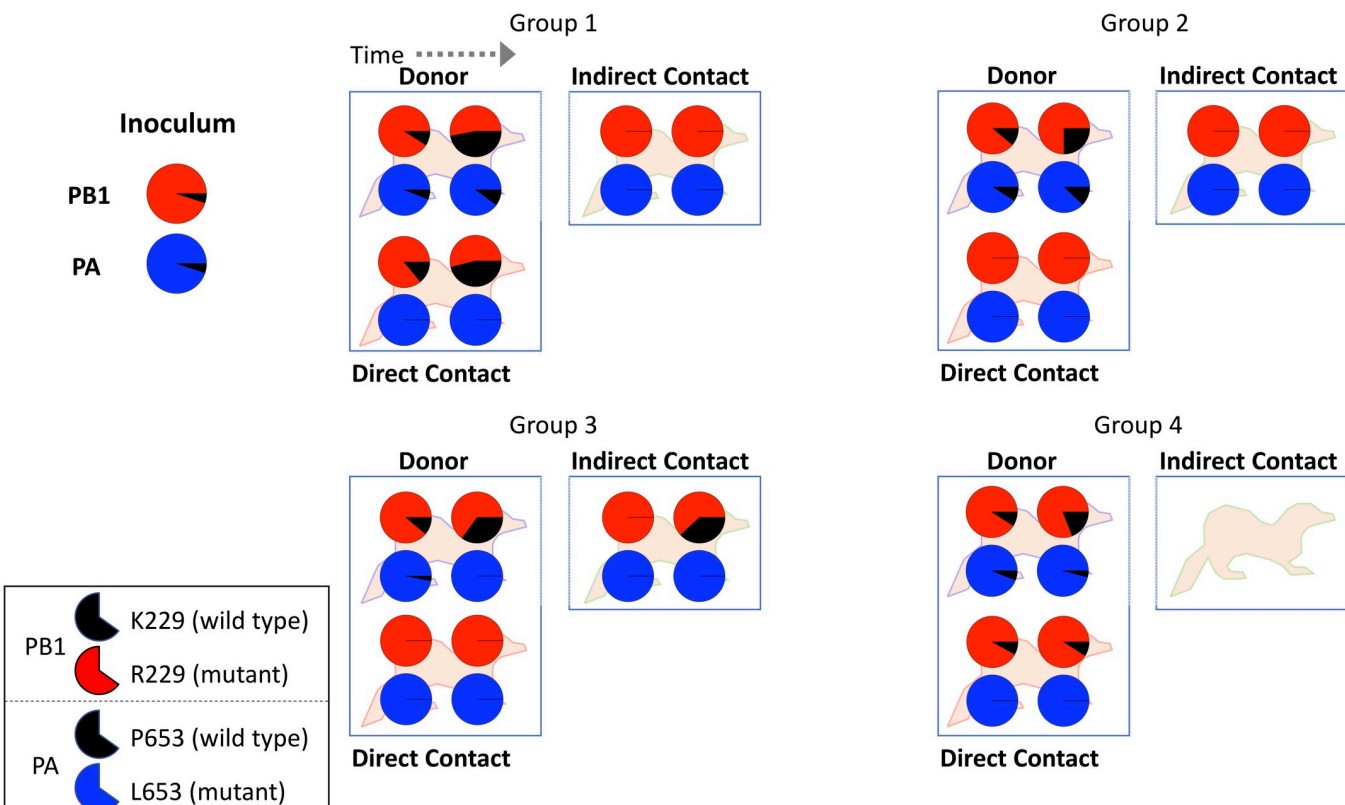

**Fig 2. Targeted sequencing of PA and PB1 using NGS showed the percentage of PB1 K229R and PA P653L mutations for donor, contact and indirect contact ferrets.** The top pie chart shows the percentage of each genotype for residue 229 in PB1 with the mutant (R229) in red and the wild type (K229) in black. The bottom pie chart shows the percentage of each genotype for residue 653 in PA with the mutant (L653) in blue and the wild type (P653) in black. The inoculum shows 5% K229 and 5% P653. For each infected ferret, two sequenced time points (as described in Fig 1) are shown. The group 4 indirect contact was not infected.

or the sentinel ferrets occurring above 5%. This confirmed that the K229R + P653L virus was able to productively infect and transmit between ferrets. There were a low number of non-synonymous mutations at >25% in direct contact (5 mutations) and respiratory sentinel ferrets (4 mutations) which were likely due to bottlenecking occurring during transmission. These mutations were not observed in multiple ferrets, which could have indicated parallel evolution and positive selection. The one exception was a mutation in M2 (T60A) which was present in both the direct and indirect contact from the same donor but in no other ferrets. This sequencing provided evidence that favipiravir resistant virus could transmit without additional compensatory mutations.

## R229 mutation selected against over time within ferrets

Next, we sought to understand how the K229R and P653L mutations might change over time within an individual ferret. In all four donor ferrets, the proportion of the PB1 wild-type amino acid, K229 increased compared to R229 from the inoculum at the earliest time point and further increased at the later time point (Fig 2). From 5% in the inoculum, K229 increased to an average of 10% in donors on day 2 and an average of 32% at the second sequencing time point (day 4 or 5). The largest increase was in donor 1 where K229 increased to 47%. By contrast, the percentage of the PA mutations showed a very different pattern in the donor ferrets, with two ferrets showing a slight increase in the wild-type PA amino acid P653, and two a

decrease in P653 on day 4/5. P653 increased slightly on average to 6% on day 2 and 6.5% on day 4/5. Sequencing of nasal wash from sentinel ferrets revealed that P653 never transmitted whereas K229 was found in 2/4 contact sentinels and 1/3 aerosol sentinels. When K229 transmitted to a sentinel ferret, the frequency of K229 increased over time in a manner similar to the donor ferret. To understand the dynamics of why K229 increased over time within a ferret but P653 did not, we created a model.

## Modelling shows a selective advantage for the P653L mutant coupled with reassortment drives genotype frequency changes

Influenza virus has a segmented genome and the key mutations in our study are located on discrete RNA segments. Next-generation sequencing does not allow detection of linkage between segments, so it is not always possible to know the exact proportion of genotypes of viruses in mixed samples. In the inoculum, PB1 K229 and PA P653 always occur together in wild-type viruses but within the donor ferrets, the evolutionary trajectory of K229 was decoupled from P653. This could have been either due to reversion of the R229 resistance mutation, which is known to have a fitness cost, or to a fitness advantage of reassortant viruses with the mutated PA segment. To understand what processes could be driving the observed genetic changes in this experiment, we constructed a simple model designed to replicate the observed virus growth dynamics within a ferret. As significant frequency changes occurred within donors between day 2 and 5 of our experiment when nasal wash titres showed no increase in viral population size, we modelled a fixed maximum population size for viruses and a fixed number of cells replenished each generation. We initially modelled infection at a high MOI (1) which allowed viruses to coinfect cells and for segments to reassort. Previous studies have shown that reassortment is common in experimental infections [32,33]. We noted that previous fitness data we generated in MDCK cells showed a large fitness disadvantage for the K229R mutant, little difference between wild type and the double mutants, and a fitness advantage for the P653L mutant [17]. The nasal cavity may have a lower temperature and so we confirmed that this fitness advantage for P653L and cost for K229R was present in MDCKs at both 37 and 33˚C (S1 Fig). Therefore, for our baseline model, we assigned a relative fitness (*f*) of 1 to wild-type virus and K229R + P653L mutant, 1.25 for the P653L mutant and 0.01 for the K229R mutant. As the actual fitness values in ferrets remain unknown, the fitness values were chosen to qualitatively reflect the fitness advantage for P653L and the fitness cost of K229R seen in MDCKs. The effect of varying these fitness values can be seen in the appendix (see S1 Appendix).

In the baseline model (Fig 3A), the initial proportions were, as in the inoculum of the experiment, 95% K229R + P653L and 5% wildtype. Reassortment due to coinfection led to the generation of K229R and P653L single mutants. The fitness advantage of the P653L mutant allowed it to outcompete both the WT and K229R + P653L. After 20 generations, the wildtype virus was lost, and the population was 100% L653. The increase of frequency in the K229 allele was representative of the dynamics seen within the donor ferrets where K229 increased from 5% to >40% in some ferrets.

Next, we tested whether the observed increase of the K229 allele was due to the fitness advantage of P653L mutant (Fig 3B) or the fitness cost to the K229R mutant (Fig 3C). We reduced the fitness of P653L such that it was equal to WT and K229R + P653L (*f* = 1). We kept other parameters the same including the fitness of K229R (0.01). Without a fitness advantage to the P653L single mutant, the WT virus was still lost after 20 generations but the frequency of P653L did not rise above 5%, the starting frequency in the inoculum (Fig 3B). The fitness cost of the K229R mutation caused the loss of the P653 allele but did not replicate the loss of the R229 allele as seen in the baseline model.

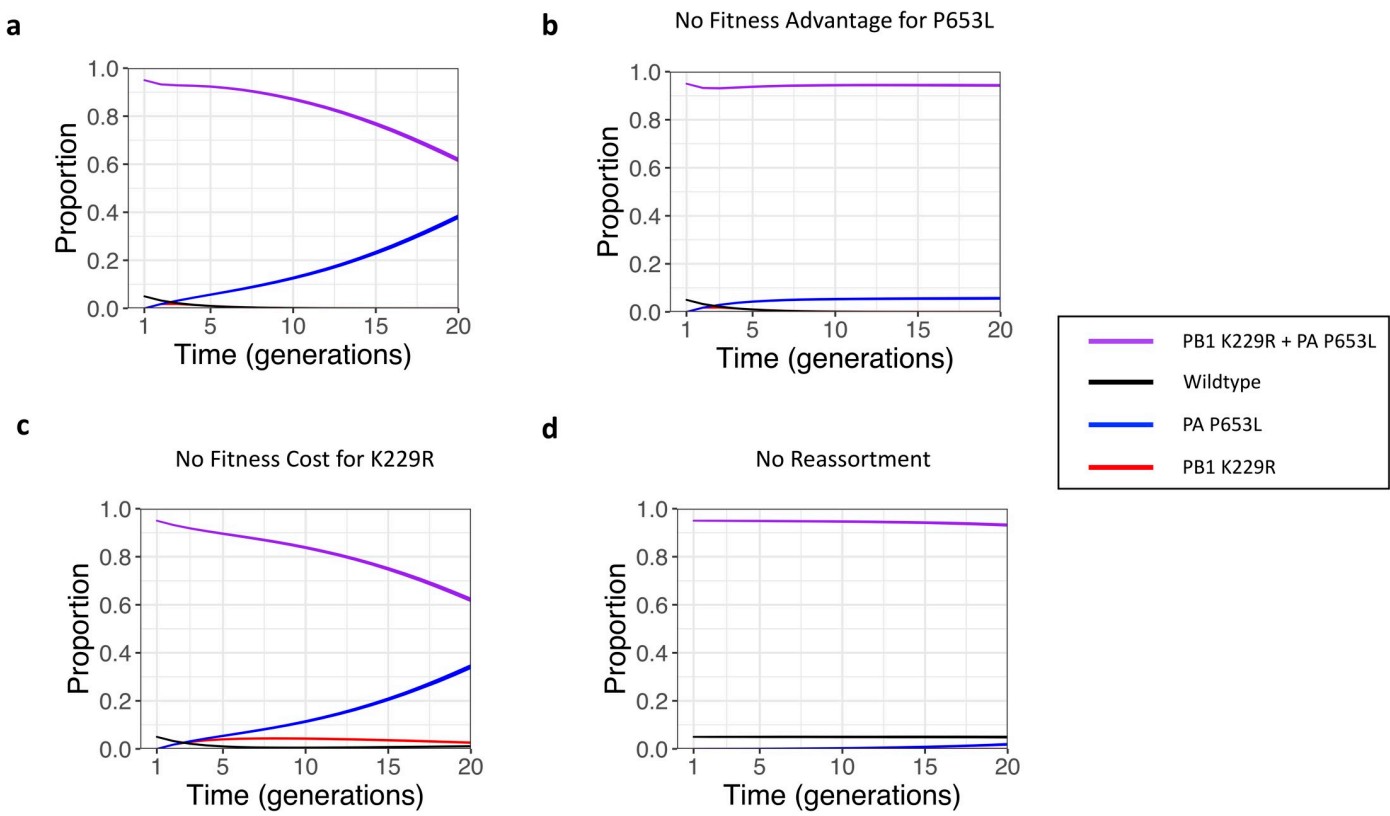

**Fig 3. A)** The proportion of each virus genotype are shown over 20 rounds of replication for a model with reassortment and mutation. The starting proportions are 5% Wild type and 95% K229R + P653L. Strain fitness for Wild type, K229R, P653L and K229R + P653L were set at 1, 0.01, 1.25 and 1 respectively. 10^6 viruses are modelled with 10^6 cells with a mutation rate, $\mu = 2 \times 10^{-4}$. **B)** As **A** but the strain fitness for Wild type, K229R, P653L and K229R + P653L were set at 1, 0.01, 1 and 1 respectively. **C)** As **A** but the strain fitness for Wild type, K229R, P653L and K229R + P653L were set at 1, 1, 1.25 and 1 respectively. **D)** As **A** but there was no reassortment allowed during coinfection, only mutation. All graphs show results from 100 replicates (the line width is from the 2.5th to the 97.5[th] percentile).

Next, we tested whether the fitness advantage of the P653L single mutant was sufficient to cause the observed increase of the K229 allele (Fig 3C). Here, we increased the fitness of K229R such that it was equal to WT and K229R + P653L ($f = 1$). We maintained a fitness advantage for P653L (1.25). Without a fitness cost to the K229R mutant, there was an increase in the P653L single mutant and a loss of WT virus. There was also a small proportion of K229R single mutant viruses which did not rise above 5%. Taken with the previous result, our model implies that when the WT virus reassorts the P653 allele is lost if there is a cost to the K229R mutant. However, the loss of the R229 allele was caused by the fitness advantage of the PA P653L single mutant not the fitness cost of the K229R mutant.

Next, we tested whether reassortment or mutation was more important for driving the dynamics of our model (Fig 3D). We removed the ability for viruses to reassort during co-infection from the model allowing genotypes to change only through mutation and selection. Without reassortment, there was very little change in genotype frequencies following 20 generations of the model. Despite the fitness advantage for P653L single mutant, the proportion of P653L did not increase above 3% after 20 generations. Mutation generated very few P653L single mutants compared to recombination. This was because co-infection with wild type and K229R + P653L, and thus the opportunity for the P653L single mutant to be generated through reassortment, occurs much more frequently than de novo mutation for the MOI and mutation rates assumed. For mutation to increase the proportion of the P653L single mutant to the level

seen in the baseline model, the mutation rate would have to be 10x higher than has been empirically measured [1] or the fitness of P653L would have to be much higher (see S1 Appendix). Therefore, our model suggested that reassortment was necessary to accurately model the observed viral dynamics seen in ferrets.

A sensitivity analysis showed that our model was robust to changes in values of the initial variables as well as changes in the model assumptions (see S1 Appendix). We showed that varying the initial proportions of WT and K229R + P653L did not lead to qualitative changes in our results nor did varying the viral load. Changing the mutation rate and MOI did not affect our results unless increased by an order of magnitude. In addition, changing the fitness cost of K229R did not quantitatively affect the results of the model.

### PA P653L does not show a fitness benefit in more recent viruses

Given the large fitness advantage of the P653L mutant virus in ferrets, it is surprising that the mutation has not been observed in pH1N1 isolates. We hypothesized that one reason the mutation might not be present is that more recent mutations in the pH1N1 virus polymerase also confer a fitness advantage, achieving the same increase in polymerase activity as P653L. Our previous work showed that the polymerase of Eng195 was less well adapted to human cells compared to later pandemic isolates and identified the N321K PA mutation as being a key mutation that led to improved polymerase activity in second and third wave pH1N1 virus isolates [34,35]. To test whether there was epistasis between N321K and P653L, we introduced the N321K mutation into Eng195 and tested polymerase using the minigenome activity (Fig 4A). Eng195 PA

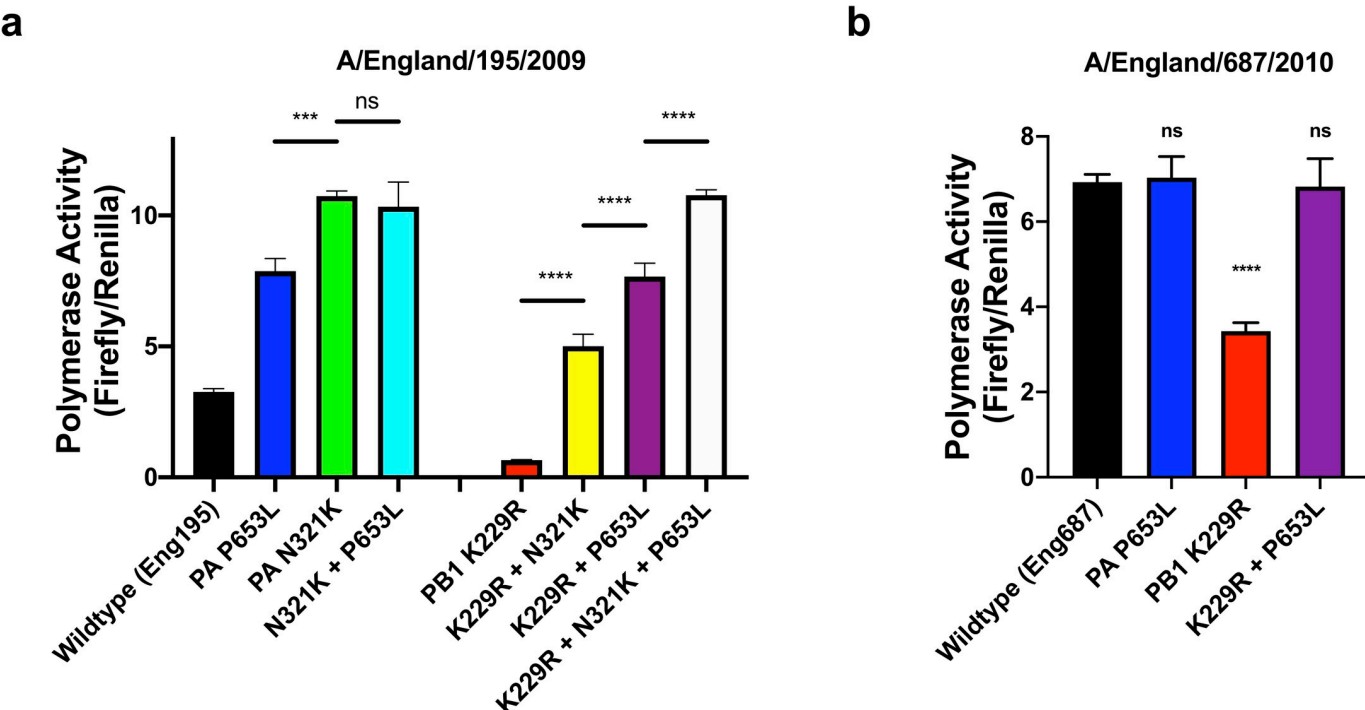

**Fig 4. Minigenome assays were performed in 293T cells in 24 or 48-well plates.** For 24 well plates, Pol I–firefly luciferase minigenome reporter, at 0.08 μg and PCAGGS-Renilla, at 0.1 μg were transfected with PCAGGS plasmids coding for wildtype and mutated polymerase subunits (PB1, PB2 and PA) and NP at 0.08, 0.08, 0.04 and 0.12 μg respectively derived from **A)** Eng195 first wave and **B)** Eng687 third wave pH1N1 virus. Plasmid amounts were halved for 48-well plate experiments. Luciferase signal was read 24 hours post-transfection. Polymerase activity is given as a ratio Firefly to Renilla signals. One-way ANOVA with Dunnett's multiple comparison test, *** p<0.001, **** p<0.0001, ns = not significant. N = 3. Error bars show s.d. Experiments were repeated 3 times and a representative experiment is shown.

N321K had higher polymerase activity compared to P653L (1-way ANOVA, p<0.001). There was no difference in polymerase activity between N321K and N321K + P653L (1-way ANOVA, p = 0.80). This implied that N321K provides a greater increase in polymerase activity than P653L and P653L provided no additional benefit to polymerase activity in the presence of N321K. We repeated this assay at 33˚C and found a similar pattern to 37˚C (S2 Fig). In order to test whether P653L provided a benefit in a N321K background, we rescued a N321K + P653L virus. Surprisingly, in MDCK cells a N321K virus showed lower growth after 24 hours than P653L or N321K + P653L (S3 Fig). This was the opposite result from our minigenome assay and previous experiments in HAEs. We do not know the reason for the loss of fitness of N321K in MDCK cells but it is reassuring that minigenome assays reflected the fitness advantage of N321K previously seen in HAEs.

Next, we wanted to test whether N321K could affect the evolution of resistance of favipiravir by compensating for the defect in polymerase activity caused by PB1 K229R. Introducing both K229R and N321K into Eng195 showed that N321K partially compensated for the loss of polymerase activity conferred by K229R but did not reach the level of polymerase activity of P653L + K229R (Fig 4A). Adding the compensatory mutation, P653L, to K229R + N321K showed full compensation of polymerase activity to the level of N321K. Next, we introduced the K229R and P653L mutations into the polymerase of a representative 3$^{rd}$ wave pandemic H1N1 virus, Eng687, which already contained N321K. As had been observed in the background of Eng195, Eng687 K229R resulted in low but appreciable polymerase activity which was fully compensated by the presence of P653L (Fig 4B). However, cost to polymerase activity caused by K229R was noticeably less in Eng687 than in Eng195. P653L only showed an increase in polymerase activity in the presence of K229R.

## Discussion

In this study, we showed that favipiravir-resistant influenza A virus could productively infect ferrets and transmit through contact transmission and via respiratory droplets. By infecting ferrets with a mix of wild-type and favipiravir resistant viruses, we sought to determine whether there was a fitness difference between the viruses. Although, the PB1 R229 and PA L653 mutations were initially present on two RNA segments within the same virus, their evolutionary trajectories were decoupled over time in the ferrets. In individual ferrets where there was a mix of K229 and R229, there was an increase in the wild-type PB1 K229 residue over time. Our modelling showed that this was not due to the fitness cost of the R229 mutation as that was adequately compensated by L653, but rather due to a fitness advantage for the P653L single mutant. This fitness advantage was implied in our previous work where the single PA mutation conferred higher polymerase activity in a minigenome assay [17] as well as the slight growth advantage we measured in the virus at 24 hours in cell culture. Despite the fitness advantage of the P653L single mutant, there was a slight increase in frequency of P653 within some ferrets. In the donor ferrets, two ferrets showed an increase and two, a decrease in the proportion of P653. We suggest that this was likely due to stochastic changes in the proportion of wild-type viruses to mutant viruses within the donor ferrets. No ferret had a frequency of P653 higher than K229 confirming that the main driver of frequency changes was the fitness advantage of the P653L single mutant. P653 did not transmit to any of the sentinel ferrets almost certainly due to low frequency of P653 and the small bottleneck size of transmission [19]. Whilst our simple model was capable of showing the rise in the P653L single mutant, it could not reproduce the stochastic changes in the frequency of P653. In our model, reassortment, coupled with the fitness cost of the K229R single mutant, led to the rapid loss of the P653 allele. The P653 allele was only maintained when there was either no reassortment or no

fitness cost to the K229R single mutant (see S1 Appendix). A spatially explicit model might have the additional complexity necessary to model the stochastic dynamics seen in the ferrets. Within the ferrets, there were likely areas within the nose infected by solely one type of virus meaning there would be reduced reassortment between the wildtype and K229R + P653L virus. The stochastic success of viruses from different areas of the nose likely drove the changes in P653 allele frequency observed within the donor ferrets.

It is notable that in 4/5 ferrets where only R229 transmitted, K229 was not generated by reversion of the K229R mutation demonstrating that reassortment was necessary for the initial production of the K229 + L653 virus and that the R229 + L653 virus was fit and capable of a productive infection. The one exception was the group 3 ferret infected through indirect contact, which showed no K229 at the first sequencing time point but 38% K229 at the second sequencing time point. Whilst this could have been caused due to reversion to K from R229, it could have also been due to subsequent reinfection from the donor (animals were exposed to donors throughout the experiment) or by outgrowth of low levels of K229, which were initially present but were not detected by sequencing. Our modelling supports that the virus with single PA mutation L653 coupled with wild-type PB1 K229 was generated by reassortment in vivo before increasing in frequency due to positive selection as seen in the schematic of the ferrets from Group 1 (Fig 5). The high levels of coinfection and reassortment necessary for such a scenario have been previously seen in experimental infections of ferrets and other animals [32,33].

Despite the large fitness advantage of L653, this mutation is not found in currently circulating pH1N1 viruses. We showed that one potential explanation for this observation is that other PA mutations have evolved that ameliorate the maladapted PA of the early pH1N1 isolates compared to more recent pH1N1 isolates. We have previously shown that later isolates from the third wave could outcompete first wave isolates in vitro due to the N321K mutation in PA [34]. Here we showed that Eng195 polymerases reconstituted with PA harbouring N321K derive no additional benefit to polymerase activity from L653. This is the most likely explanation for why L653 has not been seen in sequenced isolates. Surprisingly, PA N321K could partially compensate for the low polymerase activity of PB1 K229R despite not being structurally close to the active site (S4 Fig). This might imply that favipiravir-resistance could evolve more easily in more recent pH1N1 viruses containing PA K321 as the cost to polymerase activity of the PB1 K229R mutation is lessened by PA N321K. However, in a polymerase constellation based on the third wave virus Eng687 harbouring K229R in PB1, the P653L mutation was still required to fully compensate and to attain comparative activity to the wild-type polymerase.

A recent study examined the transmission of baloxavir resistant virus in ferrets in the absence of drug [36]. Here a single mutation PA I38T gave resistance to baloxavir. The authors found that there was a fitness cost to the PA I38T mutation which reduced transmission in competition with WT in H1N1 and H3N2 viruses [36] although in ferrets infected with 100% resistant virus, the PA I38T mutation could still transmit. The authors suggest that additional compensatory mutations would be needed for resistance to spread widely in the absence of drug. The need for compensatory mutations has been seen previously in oseltamivir resistance in H1N1, which arose in 2008 after the emergence of a permissive mutation [5]. In our previous work, we have shown that PB1 K229R causes a reduction in polymerase activity compensated by PA P653L in H1N1, H3N2 and H7N9 viruses [17]. As the compensatory mutation, P653L, was present in our experiment, the favipiravir resistant virus transmitted successfully without a clear fitness cost unlike virus resistant to baloxavir. Interestingly, Eng195 seems to suffer a greater cost to polymerase activity of K229R than more recent H1N1 viruses containing N321K, H3N2 or H7N9 [17]. A lower cost to polymerase activity of K229R might decrease

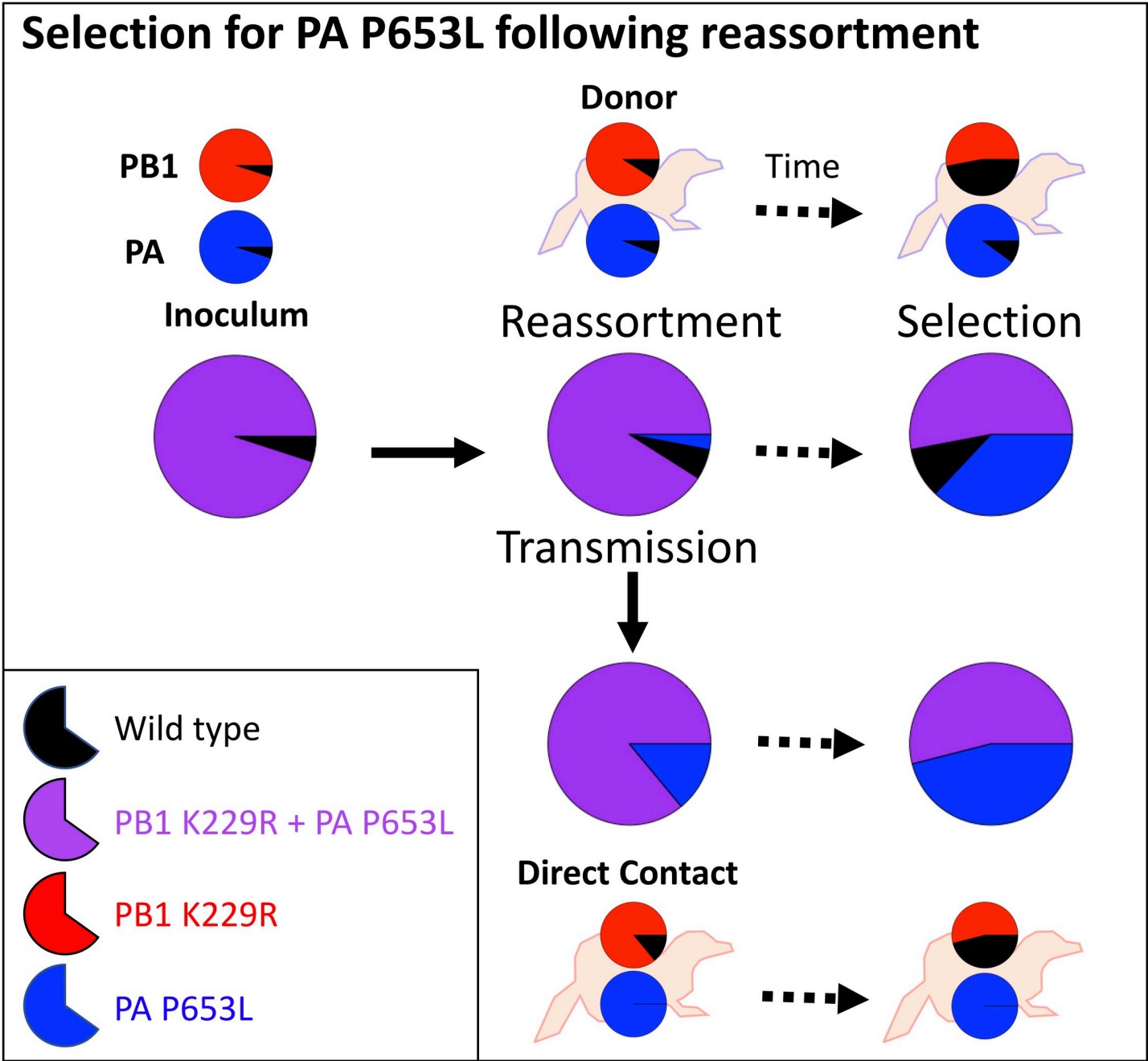

**Fig 5. Schematic explaining how virus populations change for the donor and direct contact ferrets from Group 1.** Large pie charts show the percentage of PB1 K229R + PA P653L mutant (purple) and wild-type viruses (black). Reassortment leads to the generation of the single mutant PA P653L (blue) in the donor which is transmitted to the direct contact. Smaller pie charts on each ferret show the sequencing results for PB1 and PA as in Fig 2.

the barrier to developing resistance in other influenza strains because the higher fitness of a K229R mutant might give more opportunities for a compensatory mutation to evolve. However, in all influenza A viruses tested so far, P653L is still required for full compensation of polymerase activity.

Our results have interesting implications for transmission dynamics of favipiravir resistant viruses in the absence of drug. We have demonstrated that the virus can transmit between

ferrets that are untreated with drug, which means that a localised epidemic of drug resistant virus would be possible. However, we also found that the K229R mutation was lost over time within a single ferret which implies that if the virus can be outcompeted, resistance is less likely to spread in the absence of drug. The exact probability of resistance emerging and resistance then spreading will likely depend on the precise genetic background of the virus. This has been seen in the case of oseltamivir resistance in pH1N1(2009) where, despite outbreaks of resistant virus, oseltamivir resistance remains at low levels globally [37,38]. This suggests that there may be a cost to resistance, albeit hard to measure, on this genetic background, which was not present in 2008 seasonal H1N1 [37]. In the event of a new pandemic, the specific fitness effects of both resistance mutations as well as of compensatory mutations will determine the likelihood of drug resistance emerging and spreading. Given that the compensatory mutation, P653L, is not currently found in sequenced isolates, it suggests that favipiravir resistance without a fitness cost requires multiple mutations. However, if resistance does arise, resistant viruses might continue to spread locally in the absence of drug pressure in a permissive genetic background.

## Materials and methods

### Ethics statement

All animal research described in this study was approved and carried out under a United Kingdom Home Office License, PPL 70/7501 in accordance with the approved guidelines.

### Cells and virus

Madin-Darby canine kidney (MDCK; ATCC) and HEK293T (293T) were grown in Dulbecco's modified Eagle's medium (DMEM; Invitrogen) supplemented with 10% fetal bovine serum (FBS; labtech.com), 1% penicillin-streptomycin (Invitrogen) and 1% non-essential amino acids (Gibco) at 37˚C and 5% CO2.

A/England/195/2009 (Eng195) is a first-wave isolate from the 2009 A(H1N1) pandemic grown from a reverse genetic virus [34]. Favipiravir resistant Eng195 virus (K229R + P653L) containing a K229R mutation in PB1 and a P653L mutation in PA was constructed as described previously [17].

### P653L proportions in sequenced viruses

11,455 pH1N1 (2009) viruses with full length PA segments were downloaded from GISAID on 12/03/2017. They were aligned in Geneious v11 (*Map to Reference*) to an Eng195 reference (GenBank ID: GQ166654) and all mutants at location 653 were analysed by eye.

### Animal studies

Female ferrets (20–24 weeks old) weighing 750–1000 g were acclimatized for 14 days before inoculation. Donor ferrets were lightly anaesthetized with ketamine (22 mg/kg) and xylazine (0.9 mg/kg) and then inoculated intranasally with virus diluted in phosphate buffered saline (PBS) (0.1 ml per nostril). The virus inoculum was ~10,000 plaque forming units consisting of a mix of K229R + P653L virus and Eng195 in the ratio 95:5. Sentinel ferrets were introduced day 1 post infection and remained for the duration of the experiment. Ferret body weight was measured daily to check for significant weight loss due to sickness. Ferrets were nasal washed daily, while conscious, by instilling 2 ml PBS into the nostrils, and the expectorate was collected in 250 ml centrifuge tubes. Virus titre in the nasal wash expectorate was calculated by plaque assay. The nasal wash was stored with 4% Bovine Serum Albumin Factor V (Gibco) at

-80˚C prior to RNA extraction. All sentinel ferrets were handled before donor ferrets to prevent accidental transmission of virus.

## Sequencing

Samples were chosen at multiple time points for each ferret (Fig 1) and sequenced at Public Health England. Viral RNA was extracted using easyMAG (bioMérieux) and one step Reverse-Transcription-PCR was performed with Superscript III (Invitrogen), Platinum Taq HiFi Polymerase (Thermo Fisher) and influenza specific primers. To ensure coverage of PB1 and PA, gene specific primers were used to amplify PB1 and PA which were sequenced in parallel with the whole genome samples. Sequencing libraries were prepared using Nextera library preparation kit (Illumina) and sequenced on an Illumina MiSeq generating 150-bp paired end reads. Reads were mapped with BWA v0.7.5 and converted to BAM files using SAMTools (1.1.2). Variants were called using QuasiBAM, an in-house script at Public Health England. Average depth of coverage was 12899 reads (minimum 208). Raw sequences have been deposited at https://www.ebi.ac.uk/ena (project number PRJEB39934.)

## Modelling

Viral evolution was modelled over the time using an individual-based model. The model tracked the proportion of free virions of the wild-type virus Eng195, the resistant virus K229R + P653L and viruses containing a single mutation, K229R or P653L. The model was initialised with $10^6$ virions, a mix of K229R + P653L virus and Eng195 in the ratio 95:5, as in the inoculum, and assumes a well-mixed population with virions infecting cells randomly. Evolution of the virus was tracked over 20 generations of replication, with $10^6$ cells available for infection during each generation giving an average MOI of 1. Both the number of virions and the population of cells stayed constant between generations. We allowed for varying amounts of reassortment by adjusting the ratio of viruses to cells. During each replication cycle, each virion enters a random cell. The burst size from each cell was Poisson distributed, with mean proportional to the summed fitnesses of the virion(s) infecting that cell. As default, the mean burst size for a cell infected by 1 virion was 10 [39]. In the baseline model, Eng195, K229R, P653L and K229R + P653L were assigned a relative fitness, with default values equal to 1, 0.01, 1.25 and 1 respectively. Within a cell, new segments were produced with a probability equal to the proportion of founding virions in that cell. These segments were randomly combined into virions, allowing reassortment, assuming each virus RNA segment in the cell had an equal chance, unrelated to genotype, of being incorporated into progeny virus. Newly produced virions mutated either PB1 or PA with probability $\mu = 2 \times 10^{-4}$ [1]. Excess virions were discarded randomly to maintain the viral population size.

For the model with mutation only, each cell produced whole virions for each strain, proportional to the founding virions for each strain in that cell. The newly produced virions were mutated as per above.

Code to reproduce model results can be found at https://github.com/ada-w-yan/reassortment/.

## Minigenome assay

To measure polymerase activity, pCAGGS plasmids containing genes encoding PB1, PB2, PA and NP from Eng195 or A/England/687/2010 (Eng687) were transfected into 293T cells using Lipofectamine 3000 (Thermo Fisher). Plasmids containing the K229R PB1, N321K PA and P653L PA mutations were constructed by site-directed mutagenesis. Plasmid quantities per well were PB1- 0.08 μg, PB2- 0.08 μg, PA- 0.04 μg and NP- 0.12 μg. In addition, a PolI-luc plasmid (0.08 μg), encoding a firefly luciferase minigenome reporter with influenza A segment 8 promoter sequences, was transfected with a pCAGGS-*Renilla* luciferase control (0.1 μg). After

21 hours, cells were lysed and luciferase activity measured using the Dual-Luciferase Reporter Assay kit (Promega). Polymerase activity was expressed as the ratio of Firefly:*Renilla*. Polymerase combinations were compared using 1-way ANOVA with p-values adjusted using Dunnett's Multiple Comparison test.

## Supporting information

**S1 Appendix. This appendix details the sensitivity analysis for the modelling described in Fig 3.**
(PDF)

**S1 Fig. Virus was grown on MDCK cells in 6 well plates at either 37˚C or 33˚C.** After 24 hours, the viral supernatant was plaqued and PFU/ml calculated.
(TIFF)

**S2 Fig. Minigenome assays were performed in 293T cells.** Pol I–firefly luciferase minigenome reporter, at 0.08 μg and PCAGGS-Renilla, at 0.1 μgwere transfected with PCAGGS plasmids coding for wildtype and mutated polymerase subunits (PB1, PB2 and PA) and NP at 0.08, 0.08, 0.04 and 0.12 μg respectively at 37˚C or 33˚C. Luciferase signal was read 24 hours post-transfection. Polymerase activity is given as a ratio Firefly to Renilla signals.
(TIFF)

**S3 Fig. Virus was grown on MDCK cells in 6 well plates at either 37˚C or 33˚C.** After 24 hours, the viral supernatant was plaqued and PFU/ml calculated.
(TIFF)

**S4 Fig. Structure of the bat influenza A polymerase (PDB: 4WSB).** This shows the location of the active site in white and K229 in red. The homologous position to 321 is shown in dark green and 653 in blue.
(TIFF)

## Author Contributions

**Conceptualization:** Daniel H. Goldhill, Maria Zambon, Angie Lackenby, Wendy S. Barclay.

**Data curation:** Daniel H. Goldhill, Ada Yan.

**Formal analysis:** Daniel H. Goldhill, Ada Yan, Monica Galiano.

**Funding acquisition:** Wendy S. Barclay.

**Investigation:** Daniel H. Goldhill, Ada Yan, Rebecca Frise, Jie Zhou, Jennifer Shelley, Ana Gallego Cortés, Shahjahan Miah, Omolola Akinbami, Monica Galiano.

**Methodology:** Daniel H. Goldhill, Ada Yan, Rebecca Frise, Jie Zhou.

**Supervision:** Maria Zambon, Angie Lackenby.

**Writing – original draft:** Daniel H. Goldhill, Ada Yan, Wendy S. Barclay.

**Writing – review & editing:** Daniel H. Goldhill, Ada Yan, Rebecca Frise, Jie Zhou, Jennifer Shelley, Ana Gallego Cortés, Angie Lackenby, Wendy S. Barclay.

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
