## [Decision Letter · Decision Letter 0]

7 Oct 2020

Dear Dr Goldhill,

Thank you very much for submitting your manuscript "Favipiravir-resistant influenza A virus shows potential for transmission" for consideration at PLOS Pathogens. As with all papers reviewed by the journal, your manuscript was reviewed by members of the editorial board and by several independent reviewers. In light of the reviews (below this email), we would like to invite the resubmission of a significantly-revised version that takes into account the reviewers' comments.

Based on the three reviews, there are at least a few additional in vitro studies needed to address the issue of Favipiravir-resistance in different genetic strain backgrounds and to correlate polymerase activity with replication fitness. In addition, clarity on the description and interpretation of the computational model should be added to the manuscript. While additional ferret studies would be welcome, they are not necessary to address the comments raised by the reviewers.

We cannot make any decision about publication until we have seen the revised manuscript and your response to the reviewers' comments. Your revised manuscript is also likely to be sent to reviewers for further evaluation.

Sincerely,

Seema Lakdawala, PhD

Reviews Editor

PLOS Pathogens

Sara Cherry

Section Editor

PLOS Pathogens

Kasturi Haldar

Editor-in-Chief

PLOS Pathogens

orcid.org/0000-0001-5065-158X

Michael Malim

Editor-in-Chief

PLOS Pathogens

orcid.org/0000-0002-7699-2064

Based on the three reviews, there are at least a few additional in vitro studies needed to address the issue of Favipiravir-resistance in different genetic strain backgrounds and to correlate polymerase activity with replication fitness. In addition, clarity on the description and interpretation of the computational model should be added to the manuscript.

Reviewer's Responses to Questions

**Part I - Summary**

Reviewer #1: Goldhill et al., performed transmission studies with a favipiravir-resistant strain of 2009 pandemic H1N1. The resistant virus was previously generated on the A/England/195/2009 (H1N1) background and contains the PB1 K229R and PA P653L mutations. Herein the authors evaluated the in vivo fitness effects of these resistance mutations. When the A/England virus carrying the PB1 K229R+ PA P653L mutations was inoculated into ferrets at a ratio of 95% mutant virus to 5% wild-type, the resistant virus was capable of both direct contact and airborne transmission. However, reassortant viruses carrying the PA P653L mutation with the wild-type PB1 also transmitted and eventually became the dominant strain. Subsequently, polymerase assays were performed to determine the fitness effect of the PA 653L mutation. These studies show that the PA mutation enhances polymerase activity in the A/England/195/2009 (H1N1) strain. When this mutation was introduced into a more recent pandemic H1N1 isolate, the PA P653L mutation did not confer enhanced polymerase activity. This was due to an additional mutation, N321K, that enhances polymerase activity in more recent pandemic H1N1 strains.

Reviewer #2: Goldhill et al. use an elegant combination of approaches to examine the within- and between-host dynamics of an IAV favipiravir escape mutant during mixed infection with wild type virus. In particular, a double mutant Eng195 (pH1N1) virus is examined that contains both a PB1 K229R escape mutation and a PA P653L compensatory mutation. Mixed infection with wild type virus is performed and relative proportions of the wild type and mutant segments are monitored by deep sequencing of nasal swabs from inoculated, direct contact and aerosol contact ferrets. The results reveal transmission of the escape mutant virus, but also a decline in its prevalence over time within multiple ferrets. The particular dynamics observed suggest that the PA P653L (compensatory) mutation is uncoupled from PB1 K229R by reassortment, and that this PA mutation undergoes positive selection. This model is tested in a simple but robust computational simulation; the results of which support the model given. Since the PA P653L mutation appears to be driving the dynamics observed but is not seen in circulating pH1N1 viruses, the authors then extend their thinking to recent pH1N1 viruses. Recent strains have higher baseline polymerase activity owing to a PA N321K polymorphism. In the context of PA N321K, or a full Eng687 strain background, PB1 K229R and PA P653L mutations have lesser and no effect, respectively, on pol activity. The authors conclude that the fate of 229R favipiravir resistance mutations will depend in part on the particular strain background and that resistance may arise more easily in recent pH1N1 viruses compared to 2009 pandemic isolates.

This is an interesting and elegant study, which not only gives insight into the potential for favipiravir resistance to spread but also reveals the factors governing within- and between-host dynamics in mixed IAV infection.

Reviewer #3: In this study, the authors build on their previous work, in which they showed in vitro that the favipiravir resistance mutation PB1 K229R, which conferred a fitness cost, could be compensated for by a PA P653L mutation. Here, they use a ferret model to ask whether this compensated resistance phenotype could spread in vivo, and hence whether favipiravir resistance could spread in human influenza viruses.

The paper is well conceived and for the most part well-written (other than the discussion of the model, where I confess I got a bit bogged down, see below). The experimental work is well-performed and the overall aims of the paper have significant importance. However, in my opinion the paper as written has three key weaknesses (at least two of which could be addressed without the need for further experiments).

**Part II – Major Issues: Key Experiments Required for Acceptance**

Reviewer #1: 1) The authors utilize viral polymerase activity assays to infer fitness effects of mutations in different viral strains. While these results are valid, they do not directly measure viral fitness, and the standard in the field is to perform viral growth kinetics studies to complement polymerase assays. Thus, additional in vitro growth kinetic studies of the mutant viruses are required. Moreover, the authors do not indicate the temperature at which the polymerase assays were performed. From review of the methods, this is most likely at 37ºC. Richard et al., (Nature Communications, 2020) demonstrated that during transmission between ferrets, viruses from the upper respiratory tract are transmitted. Therefore, both polymerase and growth kinetics studies should be performed at 33ºC.

Reviewer #2: none

Reviewer #3: (1) Discussion of mutations. The authors detected nucleotide-level changes but only express them as amino-acid level changes. This makes sense for the modelling sections, but when handling experimental data it means that they ignore key information that would be useful for identifying selection vs contamination or reinfection, an issue that comes up several times in the paper. Specifically, where multiple synonymous variants could give the same aa, the detection of a novel nt sequence encoding the original aa would suggest selection for that aa, in a way that detection of the original nt sequence would not. And a separate point: when describing changes in genotype frequency and detection of mutations, a statement of the depth of coverage would be helpful, as would an assessment of the intrinsic error rate of the NGS used.

(2) Which virus is interesting? The paper focusses most of its content on discussing work on A/England/195/2009 before concluding that subsequent changes in genotype render the fitness compensation constellation that had been studied irrelevant to more recent pH1N1 viruses (such as A/England/687/2010, and presumably all since). This is an important point, and the authors have my sympathy as it was probably very frustrating to discover. The reader is currently left wondering: ‘now that they’ve found this effect in a virus that is no longer circulating, what might it mean for viruses that are still out there?’ How best to address this is a balance of effort vs impact that should probably be decided in discussion with the editor. I can see that it could be ethically and financially difficult to justify repeating the ferret studies with other viruses. However, it would presumably be entirely feasible to consider the effects of fitness compensation on different backgrounds using a version of the model in Fig 3. It would also be feasible to repeat Fig 4 on a wider range of genetic backgrounds – indeed, the authors have already done much of this work in Fig 4 of their earlier study (Goldhill et al. 2018). One way or another, the paper needs to find a way to say more about viruses that people ‘care about,’ even if only indirectly.

(3) What is the model saying? I am not a specialist in modelling but I found a number of aspects of the model troubling and would like to know the author’s reasoning; I suspect most readers would too.

To begin with, it would be helpful to have a bit more background about the scenarios under which favipiravir resistance would be expected to spread ‘in the wild’ – perhaps by expanding the introductory section on the spread of resistance to previous anti-influenza drugs. Influenza is an acute infection, and favipiravir would be expected to be administered to severely ill patients only in a hospital setting. Even if the drug was used more widely, there are no reasonable circumstances in which the majority of human-to-human influenza transmission events in a human population would occur between favipiravir-treated individuals (in this sense it is very different from evasion of adaptive immunity). There are therefore a limited set of scenarios in which resistance could spread (which, to be clear, is certainly a worrying possibility), for example if the resistant constellation had little or no fitness cost in the absence of treatment, or if prophylactic treatment was excessively used in e.g. susceptible livestock. A brief but clear articulation of the selective pressures under which favipiravir resistance might be expected to spread is key context for the entire paper, but is currently missing.

Moving on to the model itself, we have Figure 3, which presents four scenarios all with starting composition 95% (K229R + P653L) to 5% WT:

(a) Relative fitness: P653L (1.25) > WT (1) = K229R+P653L (1) >> K229R (0.01). Reassortants generating K229R (653P) are lost rapidly. Reassortants generating P653L (229K) have a fitness advantage and slowly outcompete the starting WT and K229R+P653L forms. At generation 20 nearly all the population carries 653L, and 229R has declined from 95% to c.60%.

(b) Relative fitness: WT (1) = P653L (1) = K229R+P653L (1) >> K229R (0.01). Reassortants generating K229R (653P) are lost rapidly. Reassortants generating P653L (229K) are in equilibrium with K229R+P653L. WT would be in equilibrium too without reassortment but, due to the large starting amount of K229R + P653L, in any reassortant event WT is likely to mix with a double mutant and so be lost. At generation 20 nearly all the population carries 653L, and 229R remains at 95% but is now largely on a PA 653L background.

(c) Relative fitness: P653L (1.25) > WT (1) = K229R (1) = K229R+P653L (1). Reassortants generating K229R (653P) do not have a fitness disadvantage but are rare events due the starting composition; they behave much as WT in (b). Reassortants generating P653L (229K) have a fitness advantage and exponentially outcompete all other species much as in (a). At generation 20 nearly all the population carries PA 653L (slightly less than in (a)), and PB1 229R, now largely on a PA 653L background has declined from 95% to c.60%.

(d) Relative fitness: P653L (1.25) > WT (1) = K229R+P653L (1) >> K229R (0.01); no reassortment. Here the starting composition of K229R+P653L and WT is in equilibrium, and the single mutants can only be generated by spontaneous mutation at a much lower rate. Where this happens K229R (653P) is rapidly lost, and P653L (229K) once it appears begins to outcompete the other species with similar eventual kinetics as in (a) and (c) but from a much lower starting position. At generation 20 the starting composition is only slightly changed, but if you ran this out to 60 generations the eventual increase of P543L (229K) would presumably end up displacing 229R from the population.

I wouldn’t normally write this out in this much detail in a review, but I want to do so here for three reasons.

First, it took a great deal of work for me to work all of this out (most of an evening), and until I had done so I was entirely unable to make any sense of Figure 3. It should have been evident from the way the paper was written - the explanation needs to be clearer.

Second, note that in the summary above the fitness effects of K229R and P653L are strictly independent. The conclusion that ‘the fitness advantage of the PA P653L single mutant was necessary and sufficient to explain the loss of the R229 PB1 resistance mutation’ (lines 211-3) is not obvious to me. Rather, the models suggest to me that (as is intuitively obvious) K229R is lost because it carries a fitness cost, and that if a large proportion of double mutants are present in the initial mix, they will eventually dominate all reassortment events.

Third, the outputs of these models are clearly dependent on arbitrary parameters, as any model would be. The authors claim that their sensitivity analysis shows that the model outputs are ‘robust to changes in values of the initial variables’ (line 221). This is simply incorrect (which is actually reassuring, as one would expect a realistic model to be sensitive to these variables). The variable fWM, currently 1.25 (a 1.25-fold increase in fitness), behaves radically differently if increased by just 20% to 1.5. The variable fMW, currently 0.01 (a 100-fold decrease in fitness) is not tested at all, and nor is pMM (the initial proportion of double mutant virus) both of which I would argue have a major effect on the model outcomes. A much more thorough and explicit sensitivity analysis is needed, exploring model outcomes over a range of plausible values. Indeed, this could add to the value of the paper, as it would allow some consideration of what might happen in a favipiravir-treated patient where the relative fitness of WT PB1 will be, by definition, depressed, as well as considering other scenarios (see above).

**Part III – Minor Issues: Editorial and Data Presentation Modifications**

Reviewer #1: 1) In Figure 3, the authors provide a mathematical model of their findings. The model is consistent with the authors results; however, caution needs to be taken when interpreting this model. The model is derived from a single set of parameters in which ferrets were inoculated with a single mixture of viruses at a ratio of 95% mutant viruses to 5% wild-type. The model would benefit from an additional set of data inputs, as this would validate the findings. This is beyond the scope of the manuscript, but the authors should discuss the limitations of using a single set of parameters to develop the model or provide justification for why a single set of parameters is sufficient.

Reviewer #2: 1. Line 156 refers to a low number of mutations in recipient ferrets and surmises that their prevalence may be due to transmission bottlenecks. Please elaborate. Were these few mutations present at high frequencies? The prior sentence states that no mutations were found in the pol genes. Were the mutations referred to in other genes? Did their frequency increase over time in the recipient ferrets (evidence of positive selection)?

2. In the description of the baseline model (ca. line 196), a fitness value is given for only one of the four viruses. Please indicate parameters used for all viruses here. Did the values used correspond quantitatively with the data cited in ref 17? At this location in the text, it would also be helpful include basic information on how MOI was determined and what assumptions were used to govern reassortment rates - or at least refer to the appendix/methods here.

3. In the appendix:

A. Please add references indicating the sources of default parameters

B. Please include results obtained with the default parameters for comparison.

C. The default parameter for viral load seems low. Is the model sensitive to changes in this parameter?

D. How would a higher MOI impact results? (MOI may be very high within foci of infection, even in vivo.)

4. This is not absolutely necessary in my opinion, but to really round out the paper, it would be great to see a second (equivalent) ferret experiment done in the Eng687 background to test the prediction that the evolutionary dynamics of the K229R resistance mutation depends on strain background.

5. The discussion nicely discusses the results of the present study but could do more to put these results in context relative to the literature.

Reviewer #3: The arguments in the paper can often be quite hard to follow, something which would be helped considerably by a clear concluding statement at the end of each results section (currently missing from most, though not all). The discussion of Figure 3 is currently ‘upside down’ (the text refers to the frequency of K at PB1 229 while the figure shows the frequency of R).

Missing references: Line 100-2: ‘In humans so far…’ Line 302 ‘not being structurally close to the active site’ (in this case a supplementary figure would be ideal).

Line 103-4: ‘equivalent mutation’ please explain

Line 140: ‘shortly after infection and a later time point’ – please explain more clearly. It appears to be after the first increase in titre and at the last time point before the final fall in titre, but this isn’t made clear.

Line 143: ‘high levels’; line 156 ‘low number’, line 197 ‘a fitness advantage’ – please quantify (at least approximately)

Line 144 -6: ‘indicating that the K229R + P653L virus … was efficiently transmitted…’ given the (reasonable) choice to stack the deck in favour of the mutant virus the fact that it was transmitted does not actually say much about how efficient this was.

Line 156-9: if ‘bottlenecking’ could have removed low-frequency mutations, why is this not also a reasonable explanation for the loss of the WT genome segments?

Line 159: ‘pattern of repeated mutations’ meaning unclear (to me)

Line 197 (and elsewhere): I believe that the term ‘fitness’ here should more properly be ‘relative fitness’

Why is Figure 5 in the discussion rather than the results and why is only group 1 shown?

Line 303-5: ‘This might imply… cost.’ Meaning unclear (to me). Due to compensation?

Lines 322-4: ‘Given that… mutations.’ Reasoning unclear (to me).

Line 340: date of download and number of sequences downloaded required

Line 341: please briefly explain how the alignment was performed and give the accession number (or other source) of the reference sequence.

Line 385-6: ‘mean proportional to the summed fitness’ – summed (as opposed to mean) fitness implies that a cell infection with, say, 5 viruses should produce 5 times more progeny than a cell infected with one virus. Is this what was meant? If it is – given that high MOIs of IAV tend in practice to increase the rate of virus production rather than the maximum yield – does this not skew the model excessively to considering reassortant events? (It may be that in practice the nature of the Poisson distribution at MOI 1 means that the vast majority of higher order terms are negligible, but it would be good to know.)

Line 393: units for the mutation rate are missing, and there appears to be an additional superscripted term. (See also line 537.)

Line 544: how many cells for this amount of DNA?

Figure 4 legend: What is N? What measure of central tendency and variance has been plotted?

Line 557-8: Suggest removing interpretation from a figure legend.

Figure 1: the black outlines are hard to distinguish

Appendix: Keys are missing for the colours in the figures.

PLOS authors have the option to publish the peer review history of their article (what does this mean?). If published, this will include your full peer review and any attached files.

Reviewer #1: No

Reviewer #2: No

Reviewer #3: No
---

## [Editor Report · Decision Letter 1]

3 May 2021

Dear Dr Goldhill,

We are pleased to inform you that your manuscript 'Favipiravir-resistant influenza A virus shows potential for transmission' has been provisionally accepted for publication in PLOS Pathogens.

Best regards,

Seema Lakdawala, PhD

Reviews Editor

PLOS Pathogens

Sara Cherry

Section Editor

PLOS Pathogens

Kasturi Haldar

Editor-in-Chief

PLOS Pathogens

orcid.org/0000-0001-5065-158X

Michael Malim

Editor-in-Chief

PLOS Pathogens

orcid.org/0000-0002-7699-2064

The authors have successfully address the critiques raised in the first submission. In particular the expansion of the appendix to describe the sensitivity of the model was an extensive addition as was the additional experiments on polymerase activity. We are pleased to accept the revised submission with these additions.
---

## [Editor Report · Acceptance letter]

26 May 2021

Dear Dr Goldhill,

We are delighted to inform you that your manuscript, "Favipiravir-resistant influenza A virus shows potential for transmission," has been formally accepted for publication in PLOS Pathogens.

Best regards,

Kasturi Haldar

Editor-in-Chief

PLOS Pathogens

orcid.org/0000-0001-5065-158X

Michael Malim

Editor-in-Chief

PLOS Pathogens

orcid.org/0000-0002-7699-2064